# The Association between New World Alphasatellites and Bipartite Begomoviruses: Effects on Infection and Vector Transmission

**DOI:** 10.3390/pathogens10101244

**Published:** 2021-09-26

**Authors:** Angélica M. Nogueira, Monique B. Nascimento, Tarsiane M. C. Barbosa, Ayane F. F. Quadros, João Paulo A. Gomes, Anelise F. Orílio, Danielle R. Barros, Francisco Murilo Zerbini

**Affiliations:** 1Departamento de Fitopatologia, Universidade Federal de Viçosa, Viçosa 36570-900, Brazil; angelica.nogueira@ufv.br (A.M.N.); monique.nascimento@ufpel.edu.br (M.B.N.); tarsiane.barbosa@ufv.br (T.M.C.B.); ayane.quadros@ufv.br (A.F.F.Q.); joao.p.gomes@ufv.br (J.P.A.G.); anelise.orilio@ufv.br (A.F.O.); 2Instituto de Biotecnologia Aplicada à Agropecuária (BIOAGRO), Universidade Federal de Viçosa, Viçosa 36570-900, Brazil; 3Departamento de Fitossanidade, Universidade Federal de Pelotas, Capão do Leão 96160-000, Brazil; danielle.barros@ufpel.edu.br

**Keywords:** clecrusatellite, geminivirus, whitefly, natural host, experimental host, fitness

## Abstract

Begomoviruses can be found in association with alphasatellites, which are capable of autonomous replication but are dependent on the helper begomovirus for systemic infection, encapsidation and vector transmission. Previous studies suggest that the presence of NW alphasatellites (genus *Clecrusatellite*) is associated with more severe symptoms. To better understand this interaction, we investigated the effects of two alphasatellites on infectivity, symptom development, viral DNA accumulation and vector transmission of three begomoviruses in three hosts. In tomato and *Nicotiana benthamiana*, all combinations were infectious. In *Leonurus sibiricus*, only the ToYSV/ToYSA combination was infectious. The presence of EuYMA increased symptom severity of EuYMV and ToYSV in *N. benthamiana*, and the presence of ToYSA was associated with more severe symptoms of ToYSV in *N. benthamiana* and *L. sibiricus*. EuYMA increased the accumulation of ToYSV in *N. benthamiana* but reduced the accumulation of EuYMV in tomato and of ToSRV in *N. benthamiana*. The presence of ToYSA decreased the accumulation of ToYSV in *N. benthamiana* and *L. sibiricus*. ToYSA negatively affected transmission of ToSRV by *Bemisia tabaci* MEAM1. Together, our results indicate that NW alphasatellites can interact with different begomoviruses, increasing symptom severity and interfering in the transmission of the helper begomovirus. Understanding this interaction is important as it may affect the emergence of diseases caused by begomovirus–alphasatellite complexes in the field.

## 1. Introduction

Geminiviruses are plant viruses with one or two circular, single-stranded (ss) DNA genomic components, encapsidated by a single structural protein into twinned quasi-icosahedral particles. The *Geminiviridae* family is divided into multiple genera according to the type of insect vector, host range, genomic organization and phylogenetic relationships [1,2,3]. The begomoviruses, which are transmitted by whiteflies of the *Bemisia tabaci* cryptic species complex, are the most economically important geminiviruses, and cause serious crop diseases in tropical and subtropical regions of the world [4].

Begomoviruses are broadly divided into two groups, Old World (OW; Europe, Africa, Asia, and Oceania) and New World (NW; the Americas) based on genomic features and phylogeny [5,6,7]. Begomoviruses in the NW are mostly bipartite, with the two genomic components of similar size (approx. 2.6 kb) referred to as DNA-A and DNA-B. The DNA-A encodes proteins involved in viral replication, transactivation of viral genes, suppression of host defense responses and encapsidation [8,9], while the DNA-B encodes proteins associated with intra- and intercellular movement, determination of host range and suppression of defense responses [8,9,10,11]. The majority of OW begomoviruses are monopartite, with a genomic organization similar to the DNA-A of bipartite viruses, plus the presence of an additional open reading frame (ORF), which partially overlaps the CP gene, named *V2* in monopartite viruses or *AV2* in bipartite viruses [12,13].

Most OW begomoviruses are found in association with additional circular, ssDNA satellite molecules. These DNA satellites require a helper begomovirus to complete one or more steps of their infection cycle. Three types of DNA satellites have been described: alphasatellites (previously known as DNA-1), betasatellites (previously known as DNA-β), and deltasatellites [14,15]. In the NW, only alphasatellites and deltasatellites have been detected so far, mostly in association with bipartite begomoviruses [16,17,18].

Geminivirus-associated alphasatellites belong to the family *Alphasatellitidae*, subfamily *Geminialphasatellitinae* [19]. Alphasatellite genomes are approximately 1.4 kb long (half the size of begomovirus genome components), and contain a stem-loop structure, with a conserved nonanucleotide sequence (5’-TAGTATTAC-3’) comprising the origin of replication, an adenine-rich region, and a single ORF in the virion-sense strand, encoding a replication-associated protein named alpha-Rep which ensures replicational autonomy [14,19]. Alphasatellites depend on the helper virus to infect plants systemically and to be transmitted plant-to-plant by the whitefly vector [20]. The alpha-Rep protein has significant sequence identity with the master-Rep protein encoded by the DNA-R component of nanoviruses (family *Nanoviridae*), and in fact it is believed that alphasatellites evolved after a geminivirus captured one such component during co-infection of a common host [21,22].

Effects of the association of alphasatellites with helper begomoviruses are poorly understood. Early studies with OW alphasatellites (genus *Colecusatellite*) reported that they did not alter the symptoms caused by the helper begomovirus [20,22,23], and a more recent study indicated that they reduce the accumulation of the helper begomovirus [24]. Colecusatellites may be involved in pathogenicity, since their alpha-Rep protein acts as a suppressor of transcriptional gene silencing [25].

In the Americas, alphasatellites have been detected in association with bipartite begomoviruses infecting non-cultivated plants in Brazil and Cuba [16,26] and watermelon crops in Venezuela [17]. Alphasatellites were also detected in insect samples by metagenomics in Guatemala and Puerto Rico [27]. These NW alphasatellites, classified in the genus *Clecrusatellite*, are more closely related to OW alphasatellites of the genus *Ageyesisatellite* than to the initially characterized colecusatellites [19,27].

Interestingly, and unlike ageyesisatellites, which seem to attenuate the symptoms induced by the helper begomovirus [28], the clecrusatellites found in Brazil were reported to increase the severity of symptoms [16,29]. Symptoms induced by euphorbia yellow mosaic virus (EuYMV) were more severe when it was inoculated in combination with euphorbia yellow mosaic alphasatellite (EuYMA) in *Nicotiana benthamiana* and *Euphorbia heterophylla*, and the presence of the satellite was required for symptom development in *Arabidopsis thaliana* [29]. Moreover, these clecrusatellites seem to display a wide range of hosts and flexibility in their association with begomoviruses: EuYMA was detected in *E. heterophylla* (Euphorbiaceae) plants in association with EuYMV [16,29] and also in *Sida* spp. (Malvaceae) [30] associated with sida micrantha mosaic virus (SiMMV); cleome leaf crumple alphasatellite (ClLCrA) was detected in association with cleome leaf crumple virus (ClLCrV) in plants of *Cleome affinis* (Cleomaceae) [16], and tomato yellow spot alphasatellite (ToYSA) was detected in association with tomato yellow spot virus (ToYSV) infecting plants of *Leonurus sibiricus* (Lamiaceae) [30].

In this context, and considering the enormous diversity of begomoviruses infecting cultivated and non-cultivated plants in Brazil, it is important to better understand the dynamics of the interaction between NW begomoviruses and clecrusatellites, including effects on vector transmission. Mar et al. [29] reported a decrease in the transmission efficiency of EuYMV by *B. tabaci* Middle East-Asia Minor 1 (MEAM1) when EuYMA was present. It is important to determine whether this negative effect is restricted to this particular combination of begomovirus and clecrusatellite, or is a general effect of the presence of clecrusatellites in plants infected by NW bipartite begomoviruses. The objectives of this study were to investigate the effects of the interaction of EuYMA and ToYSA with three begomoviruses: EuYMV, ToYSV and tomato severe rugose virus (ToSRV) in natural and experimental hosts, and to evaluate the effect of ToYSA on whitefly transmission of tomato-infecting begomoviruses in tomato. The results indicate that effects on infectivity, symptom modulation and viral DNA accumulation vary according to the helper begomovirus, clecrusatellite and host.

## 2. Results

### 2.1. Phylogeny of Geminivirus-Associated Alphasatellites

The Bayesian-inferred tree based on full-length nucleotide sequences separated the alphasatellite isolates in seven major clusters supported by high posterior probability values and corresponding to the genera in the family (*Ageyesisatellite*, *Clecrusatellite*, *Colecusatellite*, *Draflysatellite*, *Gosmusatellite*, *Somasatellite* and *Whiflysatellite*) (Figure 1). In the cluster that corresponds to the genus *Clecrusatellite*, EuYMA and ToYSA are most closely related to tomato yellow spot alphasatellite 2, cleome leaf crumple alphasatellite and chiapas weed alphasatellite. 

### 2.2. Effects of EuYMA on Infectivity, Symptoms and Accumulation of EuYMV, ToSRV and ToYSV

The infectivity, symptom development and viral DNA accumulation of EuYMV, ToSRV and ToYSV were investigated in the presence or absence of EuYMA in tomato, a known host of both ToSRV and ToYSV, and *N. benthamiana*, a commonly used laboratory host of begomoviruses and known to be a host for the three viruses used.

In tomato, EuYMV had a lower infectivity compared to ToSRV and ToYSV, regardless of the presence of EuYMA. EuYMV was detected in 11 out of 25 plants (44%) when inoculated alone, whereas 6 out of 25 (24%) were infected with EuYMV and EuYMA (Table 1; Appendix A). The infectivity of ToSRV and ToYSV when inoculated alone was 84% (21 out of 25 plants) and 88% (22/25), respectively, and in the presence of EuYMA it was 100% for both begomoviruses (25/25) (Table 1; Appendix A). The differences in infectivity of the viruses alone or in the presence of EuYMA were not statistically significant, indicating that the presence of the EuYMA did not interfere with the infection process of these begomoviruses in this host.

The infection of tomato by EuYMV was mostly asymptomatic, with only a few plants showing faint chlorotic punctuations (Figure 2). Low detection rates of EuYMV and absence of symptoms in tomato either in the presence or absence of the alphasatellite are in agreement with the results obtained by Mar et al. [29]. Conversely, all tomato plants infected with ToSRV or ToYSV were symptomatic, and symptoms either in the presence or absence of EuYMA began to appear at 7 days post-inoculation (dpi). Symptoms in plants infected with ToSRV and EuYMA were the same compared to plants with ToSRV alone and consisted of yellow mosaic and leaf curling (Figure 2). Symptoms consisting of severe mosaic, leaf curling and leaf deformation (rugosity) were observed in ToYSV-infected plants alone or in the presence of EuYMA (Figure 2). Plants infected with ToYSV and EuYMA showed an increase in the severity of symptoms compared to plants infected with ToYSV alone (Figure 2). However, variations in the severity of symptoms were observed in both experiments independently of the presence of EuYMA. Thus, our results suggest that the presence of EuYMA is not the only factor responsible for the variation in the severity of symptoms caused by ToYSV in tomato.

In the experimental host *N. benthamiana*, the infectivity rate in plants inoculated with the virus alone was 71%, 100% and 95% for EuYMV, ToSRV and ToYSV, respectively (Table 1; Appendix A). When the plants were inoculated with EuYMV and EuYMA, the virus was detected in 95% of plants, while the alphasatellite was detected in 63% (Table 1). In the ToSRV and EuYMA combination, ToSRV was detected in 71% of the inoculated plants, while EuYMA in was detected in 67% of the plants (Table 1). The infectivity of ToYSV and EuYMA was high: ToYSV was detected in 100% of the inoculated plants and EuYMA in 95% (Table 1; Appendix A). As observed for tomato, differences in infectivity of the viruses alone or in the presence of EuYMA were not statistically significant, indicating that the presence of EuYMA did not interfere with the infection process of either begomovirus in *N. benthamiana*. 

The first symptoms in *N. benthamiana* began to appear at 5 dpi for all begomovirus/ alphasatellite combinations. Symptoms in plants infected with EuYMV alone or in the presence of EuYMA were of the same nature and consisted of mosaic and leaf deformation (as reported by Mar et al. [29]), but they were more severe in the presence of EuYMA (Figure 2). Symptoms in plants infected with ToSRV alone or in the presence of EuYMA also consisted of mosaic and leaf deformation and were of the same level of severity in presence or absence of the alphasatellite (Figure 2). In the first experiment, variations in the severity of symptoms induced by ToYSV in *N. benthamiana* (mosaic, leaf curling, leaf deformation and dwarfism) were observed regardless of the presence of EuYMA. In the second experiment, although variation in symptom severity was observed when ToYSV was inoculated alone at 14 dpi, the severity of symptoms increased in most plants in the presence of EuYMA (Appendix A). At 28 dpi, the severity of symptoms (mainly dwarfing) in plants infected with ToYSV in the presence of EuYMA was greater than in plants infected with ToYSV alone (Figure 2). Together, these results indicate that the presence of EuYMA leads to an increase in the severity of symptoms caused by EuYMV and ToYSV, but not by ToSRV, in *N. benthamiana.* However, as the ToYSV symptom severity depended on the experiment, it cannot be ruled out that uncontrolled environmental factors may partly explain the observed results.

Variation in the number of plants in which the alphasatellite was detected depending on the helper begomovirus and the host were observed, but were not statistically significant. The EuYMA detection rates in tomato and *N. benthamiana* were 56% and 95%, respectively, when associated with ToYSV and 24% and 63%, respectively, when associated with EuYMV (Table 1). When associated with ToSRV, EuYMA was detected at high rates in both hosts (72% and 67% in tomato and *N. benthamiana*, respectively) (Table 1), indicating the potential of this begomovirus/alphasatellite complex to disseminate in the field. 

To evaluate whether EuYMA affects the accumulation of EuYMV, ToSRV, and ToYSV in tomato and *N. benthamiana*, quantification of the DNA-A from each virus in the presence or absence of the alphasatellite was performed at 14 and 28 dpi. The presence of EuYMA contributed to a significant reduction in the accumulation of EuYMV in tomato plants at 28 dpi (Figure 3A; *p* = 0.0126). On the other hand, the amount of ToYSV and ToSRV in tomato plants did not present statistically significant differences when the virus was alone or in the presence of EuYMA (Figure 3A). In general, when virus and alphasatellite were inoculated together, the accumulation of EuYMA was lower compared to ToYSV and higher compared to ToSRV and EuYMV (Figure 3A), which may simply be a reflection of the high replication rate of ToYSV in tomato compared to ToSRV and EuYMV.

Contrary to what was observed in tomato plants, EuYMV accumulation did not vary in the absence or presence of EuYMA in *N. benthamiana* (Figure 3B). However, EuYMA contributed to a reduction in the accumulation of ToSRV compared to plants infected with the virus alone at 28 dpi (*p* = 0.0044) and to an increase in the accumulation of ToYSV at 14 dpi (*p* = 0.0286) (Figure 3B; Appendix A).

### 2.3. Effects of ToYSA on Infectivity, Symptoms and Accumulation of EuYMV, ToSRV and ToYSV

To determine whether ToYSA also interacts with EuYMV, ToSRV and ToYSV, the influence of this clecrusatellite on symptoms, infectivity and viral accumulation of the three begomoviruses was evaluated in tomato, *N. benthamiana* and also in *L. sibiricus,* the host from which ToYSA was originally isolated in association with ToYSV.

The overall rate of infectivity in tomato plants when the begomoviruses were inoculated alone was 55%, 59% and 60% for EuYMV, ToSRV and ToYSV, respectively (Table 2; Appendix A). When the three begomoviruses were inoculated in combination with ToYSA, the percentage of plants infected with EuYMV, ToSRV and ToYSV was 69%, 39% and 64%, respectively (Table 2). The alphasatellite was detected in 67%, 19% and 38% of the plants in which it was inoculated together with EuYMV, ToSRV and ToYSV, respectively. Differences in infectivity of the viruses alone or in the presence of ToYSA were not statistically significant, indicating that ToYSA did not interfere with the infection process of either begomovirus (Table 2; Appendix A). 

Similar to EuYMA, ToYSA also did not affect the symptoms induced by the three begomoviruses in tomato. Plants infected with EuYMV alone or in the presence of ToYSA were mostly asymptomatic, with only a few plants showing faint chlorotic punctuations (Figure 4). As with plants infected with ToSRV or ToYSV with or without EuYMA, variations in symptom severity were also observed in tomato plants infected with ToSRV or ToYSV alone or in the presence of ToYSA, consistent with the hypothesis that other factors are involved in increasing the severity of symptoms induced by these tomato infecting-begomoviruses.

*N. benthamiana* plants inoculated with EuYMV alone or in the presence of ToYSA had similar virus infectivity rates (54% and 57%, respectively; Table 2; Appendix A). ToYSA was detected in 43% of the plants. The rate of plants infected with ToSRV and ToYSV alone was 81% and 91%, respectively (Table 2). When the plants were inoculated with the two begomoviruses in the presence of ToYSA, ToSRV was detected in 70% of the plants while ToYSV was detected in 100% of the plants, differences which were not statistically significant (Table 2). ToYSA was detected in 60% and 88% of the plants in which it was inoculated together with ToSRV and ToYSV, respectively (Table 2; Appendix A). It is noteworthy that similarly high rates of detection of ToYSA and EuYMA associated with ToSRV and ToYSV were observed, indicating the efficiency of infection of both alphasatellites in this experimental host.

*N. benthamiana* plants showed the same type of symptoms induced by the three begomoviruses with or without ToYSA (Figure 4). ToYSV-infected plants in the presence of ToYSA presented more severe symptoms, including dwarfing, compared to plants infected with ToYSV alone (Figure 4; Appendix A). Some plants infected with EuYMV or ToSRV in the presence of ToYSA also developed severe symptoms; however, plants infected with the virus alone with the same severe symptoms were also observed. 

No symptoms were observed in plants of *L. sibiricus* inoculated with EuYMV or ToSRV alone or in the presence of ToYSA (data not shown). PCR analysis confirmed that none of the inoculated plants were infected with these viruses. The infectivity of ToYSV when inoculated alone was of 68% (Table 2). From a total of 30 inoculated plants with ToYSV and ToYSA, ToYSV was detected in 21 plants (70%) and ToYSA in 17 (57%) (Table 2; Appendix A). The first symptoms were observed at 6 dpi, with a mosaic that was of equivalent severity in plants inoculated with ToYSV alone or in combination with ToYSA (not shown). Interestingly, a significant increase in symptom severity was observed in the presence of ToYSA (Figure 4). In addition to mosaic, leaf distortion and blistering were verified in plants infected with ToYSV and ToYSA. Symptoms induced by ToYSV alone consisted of a severe mosaic without leaf distortion and blistering.

No significant differences in the accumulation of either begomovirus in the presence or absence of ToYSA were observed in tomato (Figure 5A). In *N. benthamiana*, no differences in the accumulation of EuYMV or ToSRV with or without ToYSA were observed (Figure 5B). Interestingly, a significant decrease in the accumulation of ToYSV in the presence of ToYSA was observed at 14 dpi (Appendix A; *p* = 0.0003). Likewise, the presence of ToYSA contributed to a reduction in the accumulation of ToYSV in *L. sibiricus* at 28 dpi (Figure 5C; *p* = 0.0379).

### 2.4. Effect of ToYSA on the Transmission of ToSRV by B. tabaci MEAM1

We evaluated the transmission of ToSRV and ToYSV in the presence or absence of ToYSA by *B. tabaci* MEAM1 in tomato plants in two independent experiments. In the first experiment, ToSRV was transmitted with 100% efficiency when inoculum source plants with ToSRV alone were used. When plants infected with ToSRV and ToYSA were used as inoculum sources, the virus was transmitted to 46% of the plants (13 out of 28) and the satellite was detected in 18% of them (Table 3). In the second experiment, ToSRV was transmitted with 88% efficiency when inoculum source plants with ToSRV alone were used, and 92% when plants infected with ToSRV and ToYSA were used as inoculum sources. However, the satellite was not detected in any plants of the second experiment (Table 3).

Symptoms in plants infected with ToSRV alone or in the presence of ToYSA consisted of mosaic, similar to the symptoms of the source plants inoculated by biolistics. Interestingly, plants inoculated with ToSRV and ToYSA, infected or not by ToYSA, recovered from symptoms in later stages of infection and showed mild mosaic or no symptoms (Appendix A). These results suggest that ToYSA interferes with the transmission of ToSRV by *B. tabaci* MEAM1 and may affect the development of symptoms induced by the virus.

ToYSV was not transmitted by *B. tabaci* MEAM1, regardless of the presence of ToYSA.

## 3. Discussion

Alphasatellites were described in the late 1990s [22] and consist of a widely diverse group of subviral infectious agents. Alphasatellites in the Old World do not seem to affect the symptoms or the viral load of the helper virus [23]. However, two reports in the New World indicate that NW alphasatellites (now classified in the genus *Clecrusatellite*) cause an increase in the symptoms induced by the helper virus, at least in some hosts [16,29]. Considering the great diversity of begomoviruses infecting cultivated and non-cultivated plants in the NW, a better understanding of the interaction dynamics between NW begomoviruses and alphasatellites of the genus *Clecrusatellite* is needed, and may provide insights into the factors that facilitate spillover events and the subsequent emergence of begomoviruses in crops. In this context, we used two clecrusatellites (EuYMA and ToYSA), three begomoviruses (EuYMV, ToSRV and ToYSV) and three hosts (*Leonurus sibiricus*, *Nicotiana benthamiana* and tomato) to study the interaction dynamics between the two types of agents. 

EuYMA and ToYSA were isolated from *Euphorbia heterophylla* and *Leonurus sibiricus* in association with EuYMV and ToYSV, respectively [16,30]. Although EuYMV and ToYSV have been sporadically detected in other hosts, including tomato [31,32,33,34], they are largely restricted to *E. heterophylla* and *L. sibiricus*, respectively. Conversely, ToSRV is one of the most widespread tomato-infecting begomoviruses in Brazil, and has an unusually wide host range that includes both non-cultivated and cultivated hosts [35,36,37,38,39]. Thus, the different combinations of begomovirus and clecrusatellite analyzed here should encompass distinct degrees of adaptation to cultivated and non-cultivated hosts. 

In the OW, a high degree of promiscuity is observed between DNA satellites (both alpha- and betasatellites) and begomoviruses [14]. This can now be extended to NW clecrusatellites, as EuYMA and ToYSA were able to interact efficiently with three begomoviruses in tomato and *N. benthamiana* following artificial inoculation. However, the occurrence of clecrusatellites is rare in the field [29,30], indicating that additional factors could be preventing a wider dissemination of these agents in nature. Thus, the actual threat posed by these agents to form novel disease complexes that could spill over to crops remains to be determined.

Our results do not indicate a significant effect of either clecrusatellite in the infectivity of the helper viruses in either host. In tomato, the low detection rate of EuYMV either in the presence or absence of EuYMA is in agreement with the results obtained by Mar et al. [29] and indicates that EuYMV is not well adapted to tomato. Nevertheless, the ubiquity of EuYMV in *E. heterophylla* and its ability to infect tomato suggest that it could spill over to this host if the vector populations are capable of efficient transmission. The fact that transmission efficiency seems to decrease in the presence of EuYMA [29] could be one reason why spillover infections of EuYMV are not common. Conversely, the high efficiencies of infection of both ToSRV and ToYSV in tomato under experimental conditions have been reported before [40,41,42,43], and our results reinforce the high adaptability of these viruses to this host, which was not affected by the presence of the satellites. Of the three hosts analyzed in this study, *N. benthamiana* had the highest infectivity rate for all three begomoviruses, alone or in the presence of EuYMA or ToYSA. This suggests that, while clecrusatellites may be promiscuous in terms of their helper begomoviruses, the interaction with host factors may be more important as far as the success of the systemic infection is concerned. 

Of all the begomovirus/clecrusatellite/host combinations analyzed in this study, an increase in symptom severity associated with the presence of the satellites was observed in *N. benthamiana* infected with EuYMV/EuYMA, ToYSV/EuYMA and ToYSV/ToYSA, and in *L. sibiricus* infected with ToYSV/ToYSA (the only combination that was infectious in this host). No differences in symptom severity were observed in tomato as a function of the presence of clecrusatellites. In a previous study, we showed that EuYMA was also associated with increased symptom severity of EuYMV in *N. benthamiana*, *Euphorbia heterophylla* and *Arabidopsis thaliana* [29]. Together, these results indicate that the presence of clecrusatellites may lead to increased symptoms in hosts to which both agents are well adapted, but not in hosts to which one or both agents may be poorly adapted. 

The increase in symptom severity observed in some combinations was associated with a decrease in the accumulation of the helper begomovirus (ToYSV/ToYSA in *N. benthamiana* and *L. sibiricus*). This suggests not only that replication of the clecrusatellite is detrimental to the helper virus (for example, by recruiting viral and/or host factors that are necessary for the helper virus to complete its infection cycle) but also that the clecrusatellite may be the agent responsible for the increased symptom severity. In this context, it is logical to assume that the alpha-Rep protein is the pathogenicity factor as it is the only protein encoded by alphasatellites. The alpha-Rep of OW alphasatellites has been reported to be a suppressor of both transcriptional and posttranscriptional gene silencing [25,44]. The EuYMA alpha-Rep is not a suppressor of posttranscriptional gene silencing [29], and it remains to be checked if the alpha-Rep of either EuYMA or ToYSA is a suppressor of transcriptional gene silencing. The presence of alphasatellites may affect begomovirus infection in more complex ways. A study with the OW colecusatellite tomato yellow leaf curl China alphasatellite (TYLCCNA), which reduces the accumulation of the begomovirus tomato yellow leaf curl China virus (TYLCCNV), showed that 27 host genes were up-regulated and 7 were down-regulated in response to TYLCCNA [24]. The authors suggest that TYLCCNA may upregulate the expression of host genes involved in viral resistance, thus reducing viral DNA accumulation during TYLCCNV infection. This is an intriguing possibility that deserves to be investigated for the NW clecrusatellites as well. 

Another interesting aspect of the clecrusatellite/begomovirus interaction is the apparent interference on vector transmission. Reduced transmission in the presence of the clecrusatellite was observed for EuYMV/EuYMA [29] and in our first experiment with ToSRV/ToYSA. Although no differences in transmission were observed in our second experiment with ToSRV/ToYSA, the clecrusatellite was not detected in the plants inoculated with both agents (even though it was detected in all inoculum source plants). Together, these results suggest that the presence of the clecrusatellite is a negative factor for vector transmission, but the differences in the two experiments performed with ToSRV/ToYSA indicate the need for further studies. Clecrusatellites (and alphasatellites in general) supposedly use the helper begomovirus capsid protein, although it is not known if the particles containing the satellite DNA are icosahedral or geminate. It is logical to assume that the presence of the clecrusatellite interferes with encapsidation of the begomoviral progeny DNA by direct competition for the capsid protein. However, such direct competition by itself should not interfere with vector transmission, as long as enough viral particles containing the begomovirus DNA are produced. A more intriguing possibility is that the presence of the clecrusatellite DNA may lead to the production of particles containing a mixture of satellite and viral DNA, which could be structurally unstable due to suboptimal DNA-CP interactions and therefore poorly transmitted by the whitefly vector. This would help to explain the results of our second experiment with ToSRV/ToYSA: only particles containing the viral DNA alone would be transmitted. Further studies on the formation of particles in plants infected by NW begomoviruses and clecrusatellites are needed to clarify this issue. Inasmuch as the nature of the interference on transmission remains unknown, it may account for the limited incidence of field plants in which clecrusatellites are detected [29,30]. 

Curiously, we were unable to transmit ToYSV using *B. tabaci* MEAM1, with or without the clecrusatellite. This maybe a peculiarity of the ToYSV isolate used for the transmission assay (BR-Bic2-99), which had been maintained in the greenhouse for approximately 5 years by successive sap inoculations when the infectious clone was obtained [37]. The amino acid sequence of the capsid protein of ToYSV-[BR-Bic2-99] has a single difference in comparison to other ToYSV isolates: a glycine instead of an aspartic acid at position 5 (data not shown). Whether this mutation was already present in the field isolate or was a consequence of the successive sap inoculations is unknown. Regardless, its lack of transmissibility by *B. tabaci* MEAM1 and MED was recently confirmed in our laboratory (C.A.D. Xavier and F.M. Zerbini, unpublished).

Our study, together with previous ones, provides evidence that the impact of alphasatellites on begomovirus infection varies according to the specific alphasatellite, host and helper begomovirus combination. An ageyesisatellite from Oman reduced betasatellite DNA accumulation, but not the accumulation of the helper begomovirus in *N. benthamiana* [28]. A colecusatellite from Mali reduced the accumulation of the helper begomovirus in the same host without attenuating symptoms [45]. The presence of the same EuYMA isolate used in our study increased symptom severity and the accumulation of EuYMV in *N. benthamiana* and *E. heterophylla* [46]. This shows that although this clecrusatellite has the ability to interact with ToYSV and ToSRV, the effect on increased viral accumulation seems to be a feature restricted to the EuYMV/EuYMA complex in these hosts. 

It is tempting to conclude that alphasatellites are secondary players in the begomovirus/DNA satellite ecosystem. However, considering the enormous diversity of begomoviruses and alphasatellites and their collective wide host range, as well as their potential to evolve, underestimating their capacity to spill over into crops and cause severe diseases would be unwise.

## 4. Materials and Methods

### 4.1. Phylogenetic Analysis

Fifty-five representative sequences of all geminivirus-associated alphasatellites including the sequences of EuYMA and ToYSA from this study and six sequences of unclassified related alphasatellites species were downloaded from GenBank. The sequences were aligned using the Muscle algorithm implemented in MEGA X [47], and phylogenetic analysis was performed using Bayesian inference in MrBayes v. 3.2.7 [48] available at the CIPRES Science Gateway [49]. The program MrModeltest v. 2.2 (https://github.com/nylander/MrModeltest2; accessed on 25 September 2021) was used to select the nucleotide substitution model using the Akaike information criterion (AIC). Two independent runs were conducted simultaneously using 50 million generations. Burn-in was set at 25% from the resulting trees. Phylogenetic trees were visualized using FigTree v. 1.4.4 (tree.bio.ed.ac.uk/software/figtree/; accessed on 25 September 2021) and edited in CorelDRAW 2019 (Corel, Ottawa, ON, Canada). The nanoalphasatellite banana bunchy top alphasatellite 1 (genus *Babusatellite*) was used as outgroup.

### 4.2. Construction of the ToYSA Infectious Clone

An infectious clone of ToYSA was constructed using the full-length clone obtained by Ferro et al. [30] from sample CF1095 of *Leonurus sibiricus* (BR-Dou1095.1-11; GenBank access number KX348228). The clone was cleaved with *Eco*RI and *Pst*I, releasing a 400 nt fragment of the satellite genome containing the origin of replication. This fragment was cloned into the pBluescript KS+ plasmid vector (Stratagene). Then, the complete copy of the satellite genome, linearized with *Eco*RI, was inserted into the "0.3mer" clone generating constructs corresponding to 1.3 copies of the genome and containing two origins of replication in the same orientation. To confirm that the ToYSA-[BR-Dou1095.1-11] 1.3mer clone was infectious, a biolistic inoculation test (item 2.3) was performed in which the clone was inoculated in plants of *L. sibiricus* together with an infectious clone of the virus with which it was originally detected (ToYSV; [30]). Infection by the two agents was assayed by polymerase chain reaction (PCR)-based amplification of genomic fragments of ToYSV and ToYSA using virus- and satellite-specific primers (Appendix A), and also by rolling-circle amplification (RCA) [50] and digestion with the same enzymes used to construct the clone. 

### 4.3. Plant Inoculations

To study the effects of the interaction between clecrusatellites and begomoviruses in natural and experimental hosts, infectious clones corresponding to DNA-A and DNA-B of the begomovirus isolates EuYMV-[BR-Cha510-10] (GenBank accession number KY559518; [29]), ToSRV-[BR-Pir1-05] (MG837738; [51]) and ToYSV-[BR-Bic2-99] (DQ336350; [40]) were inoculated alone or in the presence of EuYMA-[BR-Cha510-10] (KY559640; [29]) or ToYSA-[BR-Dou1095.1-11] by biolistics according to [52] with some modifications. Basically, ten micrograms of each viral component were mixed to 50 µL of tungsten particles (BioRad; 60 mg/mL in 50% glycerol), 50 µL CaCl_2_ (2.5 M) and 20 µL spermidine (0.1 M) for 10 min in a vortex. Then, the DNA-coated microparticles were centrifuged at 14.000× *g* for 10 s and the supernatant was discarded. The pellet was resuspended in 200 µL of 100% ethanol and centrifuged to discard the supernatant. This process was repeated twice. Finally, the pellet was resuspended in 36 µL of 100% ethanol and, aliquots of 6 µL were distributed on the carrier membranes.

For EuYMA, a total of 25 tomato (*S. lycopersicum* cv. Santa Clara) and 20 *N. benthamiana* seedlings (2–3 leaf stage, approximately 2 weeks after sowing for tomato; 2–4 leaf stage, approximately 4 weeks after sowing for tobacco) were inoculated with each begomovirus alone or in combination with the satellite in two independent experiments. For ToYSA, a total of 40 tomato plants were inoculated in three independent experiments, and a total of 25 *N. benthamiana* and 30 *L. sibiricus* plants were inoculated in two independent experiments. Healthy plants of each species inoculated with tungsten particles without DNA were used as negative controls. 

### 4.4. Detection and Quantification of Begomovirus and Alphasatellite Genomic Components

Total DNA was extracted according to the method described by Doyle and Doyle [53] from a 1-cm leaf disk of the youngest fully developed leaf of the plant at the time of collection. In the case of tomato, the apical leaflet of the youngest fully developed leaf was collected. Confirmation of infected plants was performed at 28 dpi by PCR using GoTaq Colorless Master Mix (Promega, Madison, WI, USA) and primers for the detection of the viruses and alphasatellites (Appendix A). PCR products were separated by 1% agarose gel electrophoresis and stained with ethidium bromide. After confirmation of infected plants in each treatment, 3 to 10 plants with each virus alone and with the virus in combination with each alphasatellite were selected from each independent experiment for absolute quantification of virus and satellite DNA accumulation using real-time quantitative PCR (qPCR) at 14 and 28 dpi.

A standard curve was obtained for EuYMV DNA-A, ToSRV DNA-A, ToYSV DNA-A, EuYMA and ToYSA by means of serial dilutions (5 × 10^0^ to 5 × 10^7^) of known quantities of plasmids containing one copy of the corresponding genomic component. Quantification of plasmid DNA used to construct the standard curve and of the total DNA samples were performed using a Nanodrop 2000c (ThermoFisher Scientific, Waltham, MA, USA). The primers used for quantification (0.1 µM of each primer) are listed in Appendix A. Reactions were prepared in a final volume of 10 µL using SYBR Green PCR Master Mix in a StepOnePlus Real-Time PCR System (Applied Biosystems, Waltham, MA, USA). Each sample was analyzed in triplicate by the amplification of 10 ng of total DNA. Cycling conditions consisted of an initial denaturing step of 95 °C for 10 min, followed by 40 cycles of 95 °C for 15 s and 60 °C for 1 min, with a final dissociation step to verify the specificity of amplification. Viral accumulation was determined by interpolation of the Ct values of each tested sample within the standard curve.

The data generated were initially verified for normality using the Shapiro–Wilk test [54]. Comparisons of infectivity in the presence or absence of alphasatellite in each treatment were performed using a generalized linear model (GLM, binomial family) implemented in R software. Comparisons of virus accumulation in the presence or absence of alphasatellite in each treatment were performed using a non-parametric Wilcoxon rank sum test implemented in R software [55].

### 4.5. Whitefly Transmission Assay

Tomato plants (*Solanum lycopersicum* cv. Santa Clara) biolistically inoculated with ToSRV-[BR:Pir1:05] and ToYSV-[BR:Bic2:99] with or without ToYSA were used as inoculum sources for the transmission experiments. To confirm the infection, total DNA of all plants (inoculated and healthy controls) was extracted as described by Doyle and Doyle [53] and the presence of the two agents was verified by PCR using GoTaq Colorless Master Mix (Promega, Madison, WI, USA) and primers listed in Appendix A. Two independent transmission assays were performed, both using ToSRV- and ToYSV-inoculated source plants with and without ToYSA at 28 dpi. Confirmation of infection was always performed one day before the transmission experiment. 

The whiteflies used in this study were obtained from colonies maintained in cabbage plants (*Brassica oleracea* var. *capitata*; a non-host for tomato begomoviruses) kept inside whitefly-proof screened cages in a growth chamber with a controlled temperature of 25 °C and a photoperiod of 14 h light and 10 h dark. About 1000 non-viruliferous whiteflies were transferred to a cage containing ToSRV- or ToYSV-infected tomato plants with or without ToYSA for an acquisition access period (AAP) of 48 h. After the AAP, whiteflies were randomly collected using a mouth aspirator and transferred to healthy tomato plants (30 adult insects/plant) for an inoculation access period (IAP) of 48 h. A total of thirty plants per treatment were used in each experiment. Non-viruliferous whiteflies transferred to healthy tomato plants for AAP and IAP of 48 h were used as negative controls. After the IAP, whiteflies were eliminated mechanically and by the application of acetamiprid (80 mg A.I./L). The plants were kept in a greenhouse in protected cages separated by treatment to avoid contamination. 

The appearance of symptoms was evaluated up to 28 dpi. To confirm the presence of the virus and alphasatellite in each plant, total DNA was extracted at 28 dpi (from a 1-cm leaf disk of the youngest fully developed leaf at the time of collection) and used as a template for conventional PCR as described above.

## Figures and Tables

**Figure 1 pathogens-10-01244-f001:**
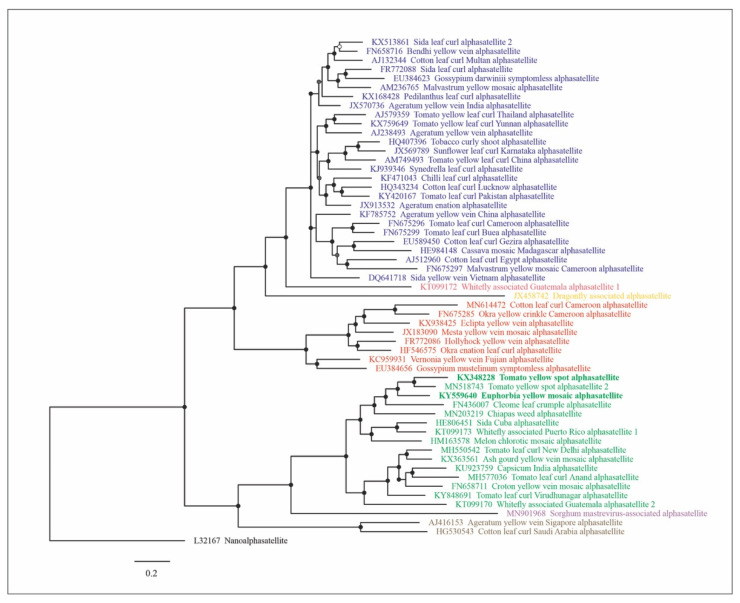
Bayesian phylogenetic tree based on representative sequences of all geminivirus-associated alphasatellites, including the sequences of EuYMA and ToYSA from this study and six sequences of unclassified related alphasatellite species. Nodes with posterior probability values lower than 0.50 are indicated by empty circles, nodes with posterior probability values between 0.50 and 0.79 are indicated by gray circles and nodes with values equal to or greater than 0.80 are indicated by filled circles. The scale bar represents the number of nucleotide substitutions per site. Isolate color indicates classification at the genus level: blue, *Colecusatellite*; coral, *Whiflysatellite*; orange, *Draflysatellite*; red, *Gosmusatellite*; green, *Clecrusatellite*; purple, *Somasatellite*; brown, *Ageyesisatellite*. The nanoalphasatellite banana bunchy top alphasatellite 1 (genus *Babusatellite*) was used as outgroup. The scale bar indicates substitutions per site.

**Figure 2 pathogens-10-01244-f002:**
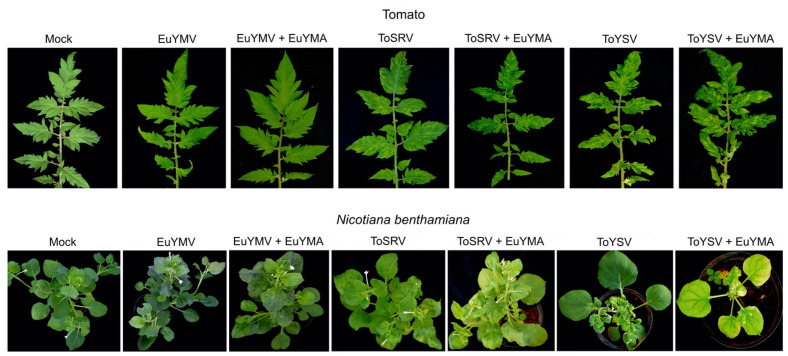
Symptoms in tomato and *N. benthamiana* plants infected with the begomoviruses euphorbia yellow mosaic virus (EuYMV), tomato severe rugose virus (ToSRV) and tomato yellow spot virus (ToYSV), in the absence or presence of euphorbia yellow mosaic alphasatellite (EuYMA) at 28 days post-inoculation.

**Figure 3 pathogens-10-01244-f003:**
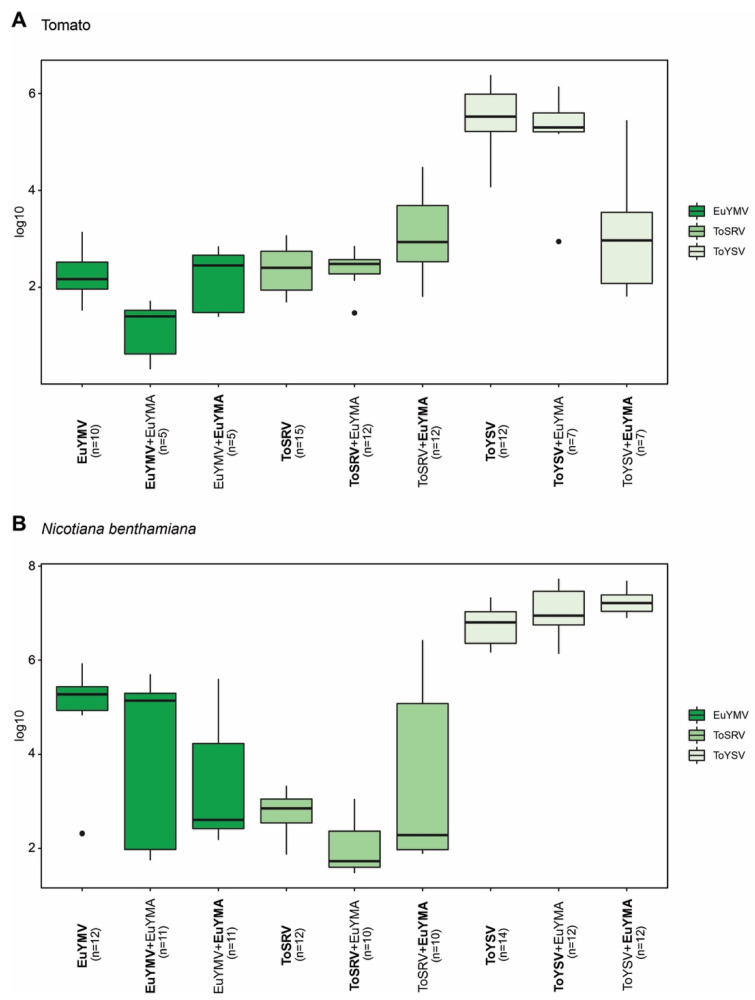
Accumulation of euphorbia yellow mosaic virus (EuYMV), tomato severe rugose virus (ToSRV) and tomato yellow spot virus (ToYSV) DNA-A in the absence or presence of euphorbia yellow mosaic alphasatellite (EuYMA). Absolute quantification of viral DNA was performed at 28 days post-inoculation in (**A**) tomato (*S. lycopersicum*) and (**B**) *N. benthamiana*. Boxplots correspond to viral accumulation presented as the log of the number of molecules. Dots indicate outlier values. The number of plants analyzed in each treatment (*n*) is indicated, and the agent (begomovirus or satellite) evaluated is in bold.

**Figure 4 pathogens-10-01244-f004:**
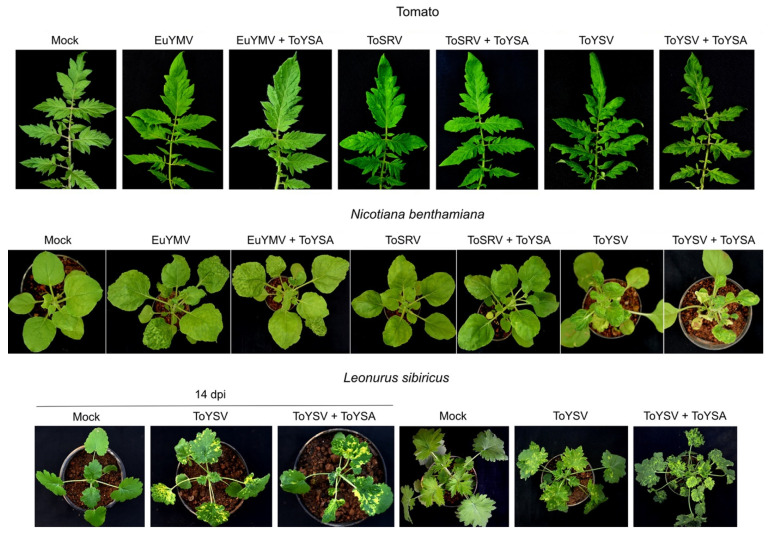
Symptoms in tomato, *N. benthamiana* and *L. sibiricus* plants infected with the begomoviruses euphorbia yellow mosaic virus (EuYMV), tomato severe rugose virus (ToSRV) and tomato yellow spot virus (ToYSV), in the absence or presence of tomato yellow spot alphasatellite (ToYSA). All images at 28 days post-inoculation, except when otherwise indicated.

**Figure 5 pathogens-10-01244-f005:**
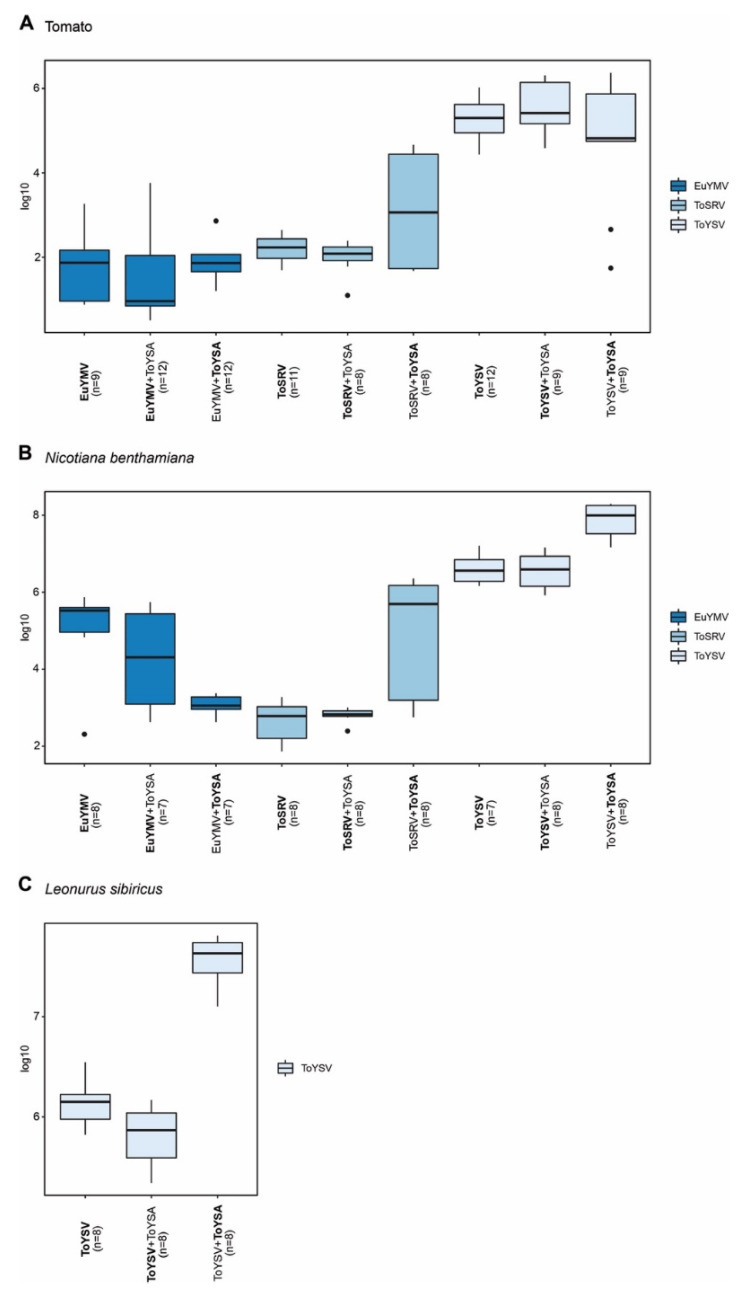
Accumulation of euphorbia yellow mosaic virus (EuYMV), tomato severe rugose virus (ToSRV) and tomato yellow spot virus (ToYSV) DNA-A in the absence or presence of tomato yellow spot alphasatellite (ToYSA). Absolute quantification of viral DNA was performed at 28 days post-inoculation in (**A**) tomato (*S. lycopersicum*), (**B**) *N. benthamiana* and (**C**) *L. sibiricus*. Boxplots correspond to viral accumulation presented as the log of the number of molecules. Dots indicate outlier values. The number of plants analyzed in each treatment (*n*) is indicated, and the agent (begomovirus or satellite) evaluated is in bold.

**Table 1 pathogens-10-01244-t001:** Infectivity of three New World bipartite begomoviruses (euphorbia yellow mosaic virus, EuYMV; tomato severe rugose virus, ToSRV; tomato yellow spot virus, ToYSV), alone or in association with euphorbia yellow mosaic alphasatellite (EuYMA), in tomato (*Solanum lycopersicum*) and *Nicotiana benthamiana*. Results correspond to the sum of two independent experiments. Results of each experiment are presented in Appendix A.

	**Tomato**					
	EuYMV		ToSRV		ToYSV	
	alone	+EuYMA	alone	+EuYMA	alone	+EuYMA
Virus detection *	11/25 ^#^ (44%)	6/25 (24%)	21/25 (84%)	25/25 (100%)	22/25 (88%)	25/25 (100%)
EuYMA detection *		6/25 (24%)		18/25 (72%)		13/25 (52%)
	** *Nicotiana benthamiana* **					
	EuYMV		ToSRV		ToYSV	
	alone	+EuYMA	alone	+EuYMA	alone	+EuYMA
Virus detection	15/21 (71%)	18/19 (95%)	18/18 (100%)	15/21 (71%)	19/20 (95%)	20/20 (100%)
EuYMA detection		12/19 (63%)		14/21 (67%)		19/20 (95%)

* Number of PCR-positive plants/number of inoculated plants confirmed at 28 days post-inoculation. ^#^ No statistically significant difference was found according to a generalized linear model test (*p* < 0.05) when each virus was alone or in the presence of EuYMA in tomato and *N. benthamiana*.

**Table 2 pathogens-10-01244-t002:** Infectivity of three New World bipartite begomoviruses (euphorbia yellow mosaic virus, EuYMV; tomato severe rugose virus, ToSRV; tomato yellow spot virus, ToYSV), alone or in association with tomato yellow spot alphasatellite (ToYSA), in tomato (*Solanum lycopersicum*), *Nicotiana benthamiana* and *Leonurus sibiricus.* Results correspond to the sum of three independent experiments in tomato and two independent experiments in *N. benthamiana* and *L. sibiricus*. Results of each experiment are presented in Appendix A.

	**Tomato**					
	EuYMV		ToSRV		ToYSV	
	alone	+ToYSA	alone	+ToYSA	alone	+ToYSA
Virus detection *	22/40 ^#^ (55%)	27/39 (69%)	23/39 (59%)	14/36 (39%)	24/40 (60%)	25/39 (64%)
ToYSA detection *		26/39 (67%)		7/36 (19%)		15/39 (38%)
	** *Nicotiana benthamiana* **					
	EuYMV		ToSRV		ToYSV	
	alone	+ToYSA	alone	+ToYSA	alone	+ToYSA
Virus detection	13/24 (54%)	13/23 (57%)	17/21 (81%)	14/20 (70%)	21/23 (91%)	25/25 (100%)
ToYSA detection		10/23 (43%)		12/20 (60%)		22/25 (88%)
	** *Leonurus sibiricus* **					
	EuYMV		ToSRV		ToYSV	
	alone	+ToYSA	alone	+ToYSA	alone	+ToYSA
Virus detection	0/23	0/29	0/24	0/25	19/28 (68%)	21/30 (70%)
ToYSA detection		n.d. ^¥^		n.d.		17/30 (57%)

* number of PCR-positive plants/number of inoculated plants confirmed at 28 days post-inoculation. ^#^ No statistically significant difference was found according to the non-parametric Wilcoxon rank sum test (*p* < 0.05) when each virus was alone or in the presence of ToYSA in tomato, *N. benthamiana* and *L. sibiricus.* ^¥^ n.d., not done.

**Table 3 pathogens-10-01244-t003:** Transmission of tomato severe rugose virus (ToSRV) and tomato yellow spot virus (ToYSV) alone or in the presence of tomato yellow spot alphasatellite (ToYSA) to tomato plants by *Bemisia tabaci* Middle East-Asia Minor 1 (MEAM1).

	Number of Infected Plants/Number of Inoculated Plants (%)				
Treatments	ToSRV *	ToSRV and ToYSA ^#^	ToSRV and ToYSA ^¥^	ToYSV ^&^	ToYSV and ToYSA ^£^
Exp. 1	30/30 (100)	13/28 (46)	5/28 (18)	0/15 (0)	0/15 (0)
Exp. 2	22/25 (88)	24/26 (92)	0/26 (0)	n.d. ^$^	n.d.

* PCR detection of ToSRV DNA-A in plants inoculated with ToSRV alone. ^#^ PCR detection of ToSRV DNA-A in plants inoculated with both ToSRV and ToYSA. ^¥^ PCR detection of ToYSA in plants inoculated with both ToSRV and ToYSA. ^&^ PCR detection of ToYSV DNA-A in plants inoculated with ToYSV alone. ^£^ PCR detection of ToYSV and ToYSA in plants inoculated with both agents. ^$^ n.d., not done.

## Data Availability

All data generated or analyzed during this study are included in this published article (and its Appendix A Files).

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
