# Peer review of "The Association between New World Alphasatellites and Bipartite Begomoviruses: Effects on Infection and Vector Transmission"

_pathogens, 2021, doi:10.3390/pathogens10101244_

Round 1

Reviewer 1 Report

The authors present an interesting study into the relationships into begomoviruses and satellites and the affect on different hosts. The paper is well written and only needs minor corrections. I have attached a PDF with highlighted comments detailing the corrections for the authors to work through.

Major Issues - There were no major issues with this manuscript. It was based on sound methods - all experiments were performed in duplicate with all virus/satellite combinations utilised in three different hosts with controls. It also builds and supports previous work in the area.

Minor Issues - The only minor issues were some typos and spelling errors. The manuscript is well written and easy to understand. All tables and Figures were well described.

Author Response

Responses to the reviewers' comments

We sincerely thank the two reviewers for their constructive and extremely helpful comments and suggestions, all of which were taken into consideration. In the responses below, reviewer comments are in regular font, and our responses are in blue italics. Line numbers refer to the revised version with track changes on.  

Reviewer 1

The authors present an interesting study into the relationships into begomoviruses and satellites and the affect on different hosts. The paper is well written and only needs minor corrections. I have attached a PDF with highlighted comments detailing the corrections for the authors to work through.

Major Issues - There were no major issues with this manuscript. It was based on sound methods - all experiments were performed in duplicate with all virus/satellite combinations utilised in three different hosts with controls. It also builds and supports previous work in the area.

Minor Issues - The only minor issues were some typos and spelling errors. The manuscript is well written and easy to understand. All tables and Figures were well described.

Thank you for such positive comments and for the corrections. All typos/spelling errors were corrected in the revised manuscript.

Reviewer 2 Report

The present work analyzes the interaction between different begomoviruses and the new world alphasatellites associated with them in different hosts raging from natural to experimental ones. As begomoviruses are one of the largest genera of plant viruses, infecting many plant species of agricultural and ecological importance, understanding the factors that determine the outcome of infection and the likelihood of emergence in new hosts is of great scientific importance. As such, the subject of this work is of interest, even more considering that the effect of these hyperparasites on begomovirus epidemics is only partially understood. The manuscript is well written and results are presented in an understandable manner. I would have just suggested a reduction in redundancy of the analyses as at times it is difficult to follow which virus-satellite combination is being analyzed, but I understand that this is somehow unavoidable. My main criticism is that results do not always support, and a stronger link between these two parts of the manuscript is needed. I list my specific comments below:

Methods

  • One of the most difficult things to interpret from this work is the statistical support for the obtained results. I am not speaking about the statistical tests, but about sample sizes. As in many occasions not every inoculated plant was infected by the virus and/or the satellite, it is not clear in how many plants were the final analyses done. I think that the authors should provide this information as “n” in the different analyses that they present. Otherwise, the reader is let to do recurrent calculations on percentages of detection over inoculated plants. Note that this difference in sample size may strongly bias the results of the statistical tests, particularly in the case of non-parametric ones (as is the case of Wilcoxon´s): for smaller sample sizes these tests are more prone to false positives.
  • It is quite difficult to interpret results on symptoms when the only measure is visual (and therefore subjective). Did the authors performed some more quantitative analysis (e.g., percentage of yellow area in the leaves, dry weight of the plant or something equivalent?). I understand that presenting just pictures is a common practice, and I would not argue against it, but a more quantitative measure (if available) would yield more solid results.

Results:

  • Section 2.2. It is not obvious to this reviewer why the authors chose a Wilcoxon test when dealing with frequencies, which would be more appropriately analyzed by chi square of Fisher´s exact tests. I don´t think the conclusions will change, but proper analyses should be presented.
  • Section 2.2. Lines 237-240. I think the authors should be cautions when stating that “Together, these results indicate that the presence of EuYMA leads to an increase in the severity of symptoms caused by EuYMV and ToYSV, but not by ToSRV, in benthamiana.”, as the ToYSV symptom severity depends on the experiment, such that it cannot be ruled out that uncontrolled environmental factors may explain the observed results.
  • Fig 3. How can virus accumulation be lower at 28 than at 14 dpi? If the purpose of the authors is to analyze the overall behavior of the virus and the satellite in the plant, the measures of their multiplication should be an average on what is going on in the whole plant. As plant viruses cannot be cleared, it is not possible that overall accumulation decreases with time. This result could be explained if only younger leaves would be sampled, as indicated for detection purposes. However, whether this was so for virus quantification is not mentioned. If this were the case, the authors should at least mention the limitations of this approach: First, it is not clear if younger leaves were collected at a similar phenological stage, which would affect the level of multiplication. Second, at 14 and 28 dpi younger leaves are at different heights of the plant and it is well known that systemic movement affects how (and which) viruses reach a given leave, which again affects the level of virus multiplication. Note also that differences in virus multiplication might be explained by younger leaves of the different treatments (with or without satellite) being sampled at different phenological stages, particularly when there are differences in symptoms, rather than by a real difference in overall virus performance.
  • Section 2.4. What would be the evolutionary advantage of interfering with the transmission of the virus for the satellite? Would not be also detrimental for the satellite that could not be further transmitted? The authors only discus on the possible mechanism of this observation, but the evolutionary implications of this result are at least as intriguing as the mechanism. Do the authors have any explanation?

Discussion

  • Overall, it seems that most cases in which the presence of the satellite has an effect are accounted by infections in benthamiana, which is clearly the “less natural” host. What is this saying about the likelihood of virus spillover in the presence of the satellite?
  • Lines 483-488. Not sure about the observed results indicating high promiscuity of NW begomoviruses. These interactions seem to be rare in field conditions according to the information provided by the authors. Therefore, although possible, there seem to be something preventing it in nature.
  • Lines 505-515. If adaptation is associated with increased symptom severity, how this can be reconciled with results indicating that most cases of increased symptoms are in benthamiana that (in theory) is a host in which none of the virus or satellites has evolved?

Author Response

Responses to the reviewers' comments

We sincerely thank the two reviewers for their constructive and extremely helpful comments and suggestions, all of which were taken into consideration. In the responses below, reviewer comments are in regular font, and our responses are in blue italics. Line numbers refer to the revised version with track changes on.  

Reviewer 2

The present work analyzes the interaction between different begomoviruses and the new world alphasatellites associated with them in different hosts raging from natural to experimental ones. As begomoviruses are one of the largest genera of plant viruses, infecting many plant species of agricultural and ecological importance, understanding the factors that determine the outcome of infection and the likelihood of emergence in new hosts is of great scientific importance. As such, the subject of this work is of interest, even more considering that the effect of these hyperparasites on begomovirus epidemics is only partially understood. The manuscript is well written and results are presented in an understandable manner. I would have just suggested a reduction in redundancy of the analyses as at times it is difficult to follow which virus-satellite combination is being analyzed, but I understand that this is somehow unavoidable. My main criticism is that results do not always support, and a stronger link between these two parts of the manuscript is needed. I list my specific comments below:

Methods

One of the most difficult things to interpret from this work is the statistical support for the obtained results. I am not speaking about the statistical tests, but about sample sizes. As in many occasions not every inoculated plant was infected by the virus and/or the satellite, it is not clear in how many plants were the final analyses. I think that the authors should provide this information as “n” in the different analyses that they present. Otherwise, the reader is let to do recurrent calculations on percentages of detection over inoculated plants. Note that this difference in sample size may strongly bias the results of the statistical tests, particularly in the case of non-parametric ones (as is the case of Wilcoxon's): for smaller sample sizes these tests are more prone to false positives.

The reviewer's point is well taken. We included the number of analyzed plants ("N") in the figures. We kept the non-parametric test for the virus quantification data, as the test is appropriate for continuous variables. As to sample size, we agree that it is small at 14 dpi, but believe that it is quite satisfactory at 28 dpi. Therefore, we kept the discussion of the 14 dpi results to a minimum, and moved the 14 dpi figures to the supplementary material. The infectivity data was reanalyzed using a generalized linear model (see below).

It is quite difficult to interpret results on symptoms when the only measure is visual (and therefore subjective). Did the authors performed some more quantitative analysis (e.g., percentage of yellow area in the leaves, dry weight of the plant or something equivalent?). I understand that presenting just pictures is a common practice, and I would not argue against it, but a more quantitative measure (if available) would yield more solid results.

We understand the reviewer's concern, but we really evaluated symptoms visually and at this point this cannot be changed. This certainly has a degree of subjectivity, but hopefully the images which are provided will be satisfactory. We can provide all images (besides the ones that were used to compose the figures) if so desired.  

Results:

Section 2.2. It is not obvious to this reviewer why the authors chose a Wilcoxon test when dealing with frequencies, which would be more appropriately analyzed by chi square of Fisher´s exact tests. I don´t think the conclusions will change, but proper analyses should be presented.

Thank you for this comment. We reanalyzed the infectivity data using a generalized linear model, which is more appropriate for discrete variables and variable sample sizes. Nevertheless, the results did not change.

Section 2.2. Lines 237-240. I think the authors should be cautions when stating that “Together, these results indicate that the presence of EuYMA leads to an increase in the severity of symptoms caused by EuYMV and ToYSV, but not by ToSRV, in benthamiana.”, as the ToYSV symptom severity depends on the experiment, such that it cannot be ruled out that uncontrolled environmental factors may explain the observed results.

The reviewer is correct. We added a sentence to this effect to the revised manuscript (L 244-246).

Fig 3. How can virus accumulation be lower at 28 than at 14 dpi? If the purpose of the authors is to analyze the overall behavior of the virus and the satellite in the plant, the measures of their multiplication should be an average on what is going on in the whole plant. As plant viruses cannot be cleared, it is not possible that overall accumulation decreases with time. This result could be explained if only younger leaves would be sampled, as indicated for detection purposes. However, whether this was so for virus quantification is not mentioned. If this were the case, the authors should at least mention the limitations of this approach: First, it is not clear if younger leaves were collected at a similar phenological stage, which would affect the level of multiplication. Second, at 14 and 28 dpi younger leaves are at different heights of the plant and it is well known that systemic movement affects how (and which) viruses reach a given leave, which again affects the level of virus multiplication. Note also that differences in virus multiplication might be explained by younger leaves of the different treatments (with or without satellite) being sampled at different phenological stages, particularly when there are differences in symptoms, rather than by a real difference in overall virus performance.

We always collected the youngest, fully developed leaf of the plant, at both 14 and 28 dpi. It is true that plants were higher at the second time point, but the fact that symptoms were clearly visible in these young leaves at both time points would argue that systemic movement is not impaired at 28 dpi. Although we do not have a detailed time course of virus accumulation for these specific virus/host combinations, we believe the lower accumulation at 28 dpi compared to 14 dpi is simply due to a normal decrease on virus load over time. Incidentally, we showed in a recent publication (Pinto et al., Virus Res 292:198234, 2021) that the number of HTS reads that mapped to ToSRV in tomato and Nicandra physaloides decreased over time (we analyzed three time points: 30, 75 and 120 dpi). 

Section 2.4. What would be the evolutionary advantage of interfering with the transmission of the virus for the satellite? Would not be also detrimental for the satellite that could not be further transmitted? The authors only discus on the possible mechanism of this observation, but the evolutionary implications of this result are at least as intriguing as the mechanism. Do the authors have any explanation?

Thank you for this comment. We do not have an explanation, but could provide some "educated speculation" (L 561-570).

Discussion

Overall, it seems that most cases in which the presence of the satellite has an effect are accounted by infections in benthamiana, which is clearly the “less natural” host. What is this saying about the likelihood of virus spillover in the presence of the satellite?

Actually, we observed increased symptom severity also in L. sibiricus, and in a previous publication (Mar et al., J Gen Virol 2017), also reported it in Euphorbia heterophylla and Arabidopsis thaliana.

Lines 483-488. Not sure about the observed results indicating high promiscuity of NW begomoviruses. These interactions seem to be rare in field conditions according to the information provided by the authors. Therefore, although possible, there seem to be something preventing it in nature.

Good point. It is definitely true that the occurrence of clecrusatellites is rare in the field. So we agree with the reviewer that something could be preventing a wider dissemination of these agents in nature. Nevertheless, they were able to interact with three begomoviruses in two hosts when inoculated artificially. So the capacity to interact with different helper viruses (i.e., promiscuity) exists. We modified the text accordingly (L 490-499). 

Lines 505-515. If adaptation is associated with increased symptom severity, how this can be reconciled with results indicating that most cases of increased symptoms are in benthamiana that (in theory) is a host in which none of the virus or satellites has evolved?

Please see the answer to the first comment on the Discussion. We also observed increased symptom severity in natural hosts.